# Carboplatin Enhances the Activity of Human Transient Receptor Potential Ankyrin 1 through the Cyclic AMP-Protein Kinase A-A-Kinase Anchoring Protein (AKAP) Pathways

**DOI:** 10.3390/ijms20133271

**Published:** 2019-07-03

**Authors:** Kanako Miyano, Seiji Shiraishi, Koichiro Minami, Yuka Sudo, Masami Suzuki, Toru Yokoyama, Kiyoshi Terawaki, Miki Nonaka, Hiroaki Murata, Yoshikazu Higami, Yasuhito Uezono

**Affiliations:** 1Division of Cancer Pathophysiology, National Cancer Research Institute, Tokyo 104-0045, Japan; 2Emergency Life-Saving Technique Academy of Tokyo (ELSTA TOKYO), Hachioji-shi, Tokyo 192-0364, Japan; 3Department of Anesthesiology and Critical Care Medicine, Jichi Medical University, Tochigi 329-0498, Japan; 4Laboratory of Molecular Pathology and Metabolic Disease, Faculty of Pharmaceutical Sciences, Tokyo University of Science, Chiba 278-8510, Japan; 5Department of Anesthesiology, Nagasaki University Graduate School of Biomedical Sciences, Nagasaki 852-8501, Japan; 6Translational Research Center, Research Institute of Science and Technology, Tokyo University of Science, Chiba 278-8510, Japan; 7Division of Supportive Care Research, Exploratory Oncology Research and Clinical Trial Center, National Cancer Center, Tokyo 104-0045, Japan; 8Innovation Center for Supportive, Palliative and Psychosocial Care, National Cancer Center, Tokyo 104-0045, Japan; 9Department of Comprehensive Oncology, Nagasaki University Graduate School of Biomedical Sciences, Nagasaki 852-8523, Japan

**Keywords:** carboplatin, CIPN, TRPA1, cAMP, PKA, AKAP

## Abstract

Carboplatin, an anticancer drug, often causes chemotherapy-induced peripheral neuropathy (PN). Transient receptor potential ankyrin 1 (TRPA1), a non-selective cation channel, is a polymodal nociceptor expressed in sensory neurons. TRPA1 is not only involved in pain transmission, but also in allodynia or hyperalgesia development. However, the effects of TRPA1 on carboplatin-induced PN is unclear. We revealed that carboplatin induced mechanical allodynia and cold hyperalgesia, and the pains observed in carboplatin-induced PN models were significantly suppressed by the TRPA1 antagonist HC-030031 without a change in the level of TRPA1 protein. In cells expressing human TRPA, carboplatin had no effects on changes in intracellular Ca^2+^ concentration ([Ca^2+^]_i_); however, carboplatin pretreatment enhanced the increase in [Ca^2+^]_i_ induced by the TRPA1 agonist, allyl isothiocyanate (AITC). These effects were suppressed by an inhibitor of protein kinase A (PKA). The PKA activator forskolin enhanced AITC-induced increase in [Ca^2+^]_i_ and carboplatin itself increased intracellular cyclic adenosine monophosphate (cAMP) levels. Moreover, inhibition of A-kinase anchoring protein (AKAP) significantly decreased the carboplatin-induced enhancement of [Ca^2+^]_i_ induced by AITC and improved carboplatin-induced mechanical allodynia and cold hyperalgesia. These results suggested that carboplatin induced mechanical allodynia and cold hyperalgesia by increasing sensitivity to TRPA1 via the cAMP-PKA-AKAP pathway.

## 1. Introduction

Carboplatin, a platinum-based anticancer drug, is widely used as a chemotherapy for various types of tumors, including ovarian cancer, cervical cancer, non-small-cell lung cancer, and malignant lymphoma [1,2,3]. In cancer cells, carboplatin crosslinks DNA and inhibits its replication. However, carboplatin is neurotoxic to sensory neurons, where it primarily damages dorsal root ganglia (DRG) and leads to development of chemotherapy-induced peripheral neuropathy (CIPN), such as allodynia and hyperalgesia [4,5,6,7,8]. However, there are few ways to effectively prevent and palliate CIPN [9]. In addition, these drugs not only decrease the quality of life of cancer patients, but also led patients to change or abort cancer chemotherapy, implicating serious problems for the prognosis of cancer patients [9].

Transient receptor potential ankyrin 1 (TRPA1) is a polymodal nociceptor (pain sensor) expressed in primary sensory neurons (DRG neurons) and involved in pain transmission [10,11,12,13,14]. TRPA1 is activated by noxious stimulus, including cold temperature (<17 °C), alkaline condition, and inflammatory mediators, such as oxidative stress [10,11,12,14]. Many studies have indicated that TRPA1 is also involved in allodynia and hyperalgesia development [15,16,17,18,19,20,21,22,23,24,25,26,27,28]. Moreover, recent reports have shown that TRPA1 plays an important role in CIPN caused by cisplatin or oxaliplatin, both of which are platinum-based drugs [16,18,19,20,21,22,23,24,25,26,27,28]. However, the role of TRPA1 in carboplatin-induced peripheral neuropathy is unclear. Therefore, this study aimed to reveal the pathogenic mechanisms of carboplatin-induced peripheral neuropathy. We analyzed the effects of an TRPA1 inhibitor on carboplatin-induced pain behavior, and measured the level of TRPA1 protein in DRG using carboplatin-induced peripheral neuropathy model mice. Furthermore, we investigated the mechanisms of effects of carboplatin on the activity or sensitivity to TRPA1 using human TRPA1 (hTRPA1)-expressing human embryonic kidney 293 (HEK293) cells.

## 2. Results

### 2.1. Carboplatin-Treated Mice Developed Mechanical and Cold Allodynia through Activation of TRPA1

We examined the effects of carboplatin on mechanical allodynia, as well as cold and heat hyperalgesia in mice. As shown in Figure 1A, carboplatin significantly decreased the paw withdrawal threshold for mechanical stimulus from Day 3, indicating that carboplatin induced mechanical allodynia. In the acetone test, carboplatin increased responses of paw flinching and biting from Day 3, and significant increases in such response was observed at Day 3 and 7, showing that carboplatin induced cold hyperalgesia (Figure 1B). However, carboplatin did not alter the paw withdrawal threshold for heat stimulus and body weight (Figure 1C,D).

To elucidate the role of TRPA1 activation in carboplatin-induced peripheral neuropathy, we examined the effects of a TRPA1 inhibitor on both mechanical allodynia and cold hyperalgesia induced by carboplatin. The TRPA1 antagonist HC-030031 at 30 mg/kg significantly improved the paw withdrawal threshold for mechanical stimulus at 30 and 60 min in a time-dependent manner (Figure 2A). At 60 min after injection of HC-030031, 30 and 100 mg/kg HC-030031 significantly improved mechanical allodynia in a dose-dependent manner (Figure 2B). Moreover, HC-030031 (30 mg/kg) also significantly suppressed cold hyperalgesia at 30 min (Figure 2C,D).

### 2.2. Carboplatin did not Increase the level of TRPA1 Protein in the Lumbar DRG of Carboplatin-Induced Peripheral Neuropathy Model Mice

To clarify how TRPA1 is related to carboplatin-induced peripheral neuropathy, we first investigated the amount of TRPA1 protein in the DRG of model mice. Compared with control mice, carboplatin-treated mice did not show a change in the protein level of TRPA1 in DRG (Figure 3).

### 2.3. Carboplatin Enhanced TRPA1 Activation in a Time- and Dose-Dependent Manner

Some reports have shown that the activity of DRG neurons expressing TRP channels was enhanced in CIPN model animals [16,18,19,20,21,22,23,24,25,26,27,28,29,30,31,32,33]. Therefore, we investigated the effects of carboplatin on TRPA1 activation in hTRPA1-expressing HEK293 cells using a Ca^2+^ imaging assay. As shown in Figure 4A, treatment with carboplatin alone (gray line) did not induce changes in intracellular Ca^2+^ concentration ([Ca^2+^]_i_). Pretreatment with carboplatin enhanced the increase in [Ca^2+^]_i_ induced by allyl isothiocyanate (AITC), a TRPA1 agonist, in a time-dependent manner (black line) (Figure 4A). Quantified data showed that pretreatment with carboplatin for 30 and 60 min, significantly enhanced the AITC-induced increases in [Ca^2+^]_i_ (Figure 4B). Pretreatment of carboplatin (1, 10, 100 µM) for 30 min enhanced the AITC-induced increase in [Ca^2+^]_i_ in a dose-dependent manner (Figure 4C). Quantified data showed that 10 and 100 µM of carboplatin significantly enhanced the AITC-induced increase in [Ca^2+^]_i_ (Figure 4D).

### 2.4. Involvement of Cyclic Adenosine Monophosphate (cAMP)-Protein Kinase A (PKA) in Carboplatin-Induced Enhancement of TRPA1 Activation

Many reports have shown that phospholipase C (PLC) and protein kinase A (PKA) played important roles in the enhancement of sensitivity to TRPA1 [34,35]. To reveal the intracellular signaling mechanism of carboplatin on sensitivity to TRPA1, we investigated the effects of inhibitors of PLC and PKA on carboplatin-induced enhancement of TRPA1 activation in hTRPA1-expressing HEK293 cells. As shown in Figure 5A, the PLC inhibitor U73122 (1 µM) did not significantly affect carboplatin-induced enhancement of TRPA1 activation (Figure 5A). KT5720 (0.1 µM), a PKA inhibitor, significantly decreased carboplatin-induced increase in TRPA1 activation (Figure 5B). In addition, the PKA activator forskolin (10 µM) significantly enhanced AITC-induced increase in [Ca^2+^]_i_ (Figure 5A). Therefore, we examined the effects of carboplatin on the amount of intracellular cAMP in hTRPA1-expressing HEK293 cells. Compared with the vehicle, carboplatin (10 µM) significantly increased in the amount of intracellular cAMP at 30 min (Figure 5B).

### 2.5. Involvement of A-Kinase Anchoring Protein (AKAP) in TRPA1 Activation or Carboplatin-Induced Peripheral Neuropathy

It is well known that PKA phosphorylates target proteins via AKAP [36,37,38,39,40,41]. We investigated the effects of an AKAP inhibitor on TRPA1 activation in hTRPA1-expressing HEK293 cells, particularly on mechanical allodynia and cold hyperalgesia in carboplatin-induced peripheral neuropathy. In hTRPA1-expressing HEK293 cells, AKAP St-Ht31 inhibitor peptide (AKAP I; 50 µM) significantly decreased carboplatin-induced increases in TRPA1 activation (Figure 6A). Intrathecal injection of AKAP I (20 µg) for 60 min significantly improved the paw withdrawal threshold for mechanical stimulus in carboplatin-induced peripheral neuropathy model mice (Figure 6B). In addition, intrathecal injection of AKAP I (20 µg) for 30 min significantly decreased cold hyperalgesia (Figure 6C).

## 3. Discussion

Our present study revealed that carboplatin induced both mechanical allodynia and cold hyperalgesia via TRPA1 activation (Figure 1 and Figure 2). However, carboplatin did not alter the amount of TRPA1 protein in the DRG of model mice (Figure 3). Furthermore, carboplatin enhanced AITC-induced TRPA1 activation, although carboplatin alone did not activate TRPA1 (Figure 4). These enhancements were involved in the cAMP-PKA-AKAP pathway (Figure 4, Figure 5 and Figure 6). Moreover, it was indicated that an AKAP inhibitor significantly improved carboplatin-induced mechanical allodynia and cold hyperalgesia (Figure 6). Taken together, these data suggested that both mechanical allodynia and cold hyperalgesia induced by carboplatin were induced by the enhancement of sensitivity to TRPA1 via the cAMP-PKA-AKAP pathway.

Platinum-based anticancer drugs include cisplatin and oxaliplatin, in addition to carboplatin. These drugs are accumulated in DRG neurons, thereby causing damage to neurons [16]. However, the types of pain induced in CIPN model animals by each of these drugs are different [19,21,22,23,24,25,27,28,29,30,31,32,33]. In the present study, carboplatin induced mechanical allodynia and cold hyperalgesia in mice (Figure 1), both of which were significantly suppressed by a TRPA1 inhibitor, HC-030031 (Figure 2). Many reports have shown that oxaliplatin induces both cold hyperalgesia and mechanical allodynia through TRPA1 activation [18,19,21,22,23,24,25,27,33]. On the contrary, cisplatin induces not only mechanical allodynia and cold hyperalgesia but also heat hyperalgesia via other TRP channels, such as TRPV1, instead of TRPA1 [18,21,27,28,29,30,31,32]. Taken together, these reports suggested that carboplatin was more similar to oxaliplatin than to cisplatin, although further studies are needed to clarify such difference occurs.

In our present study, the expression level of TRPA1 protein did not change in carboplatin-induced CIPN model mice (Figure 3), and carboplatin enhanced AITC-induced TRPA1 activation in hTRPA1-expressing HEK293 cells (Figure 4). Increases in the levels of TRPA1 protein in the DRG neurons of CIPN model animals induced by cisplatin or oxaliplatin have been reported [22,23,24,27,31,42,43], but there were conflicting reports indicating no changes in the levels of this protein [16,18,19,25,26,27]. These conflicting reports had shown that sensitization to TRPA1 occurred instead of increased level of TRPA1 protein. Taken together, these reports suggested that carboplatin-induced CIPN was induced by changes in sensitization to TRPA1.

Many reports have indicated that PKA played an important role in sensitization to TRPA1 [34,35]. In this study, carboplatin-induced enhancement of TRPA1 activation was significantly suppressed by the PKA inhibitor KT5720 (Figure 5A). Carboplatin increased the amount of intracellular cAMP in hTRPA1-expressing HEK293 cells (Figure 5B), which then activated PKA. Anand U. et al. [16] have shown that oxaliplatin, in association with via neuronal damage, induces an increase in intracellular cAMP and then induces sensitization of rat DRG neurons to TRP. Although further studies are needed to elucidate how carboplatin increases intracellular cAMP, these data suggested that carboplatin-activated cAMP-PKA pathway triggered an increase in sensitivity to TRPA1.

The present study showed that a carboplatin-induced increase in sensitivity to TRPA1 was significantly decreased by not only a PKA inhibitor, but also an AKAP inhibitor (Figure 5 and Figure 6). AKAP, a scaffolding protein, is required for PKA activation, and it enhances the sensitization of several receptors and ion channels via PKA [38,39,40,41]. Brackley et al. [41] have reported that AKAP facilitates phosphorylation of TRPA1, and increases the sensitivity of DRG neurons to TRPA1. These data suggested that carboplatin-induced sensitization to TRPA1 was induced by phosphorylation of TRPA1 through the cAMP-PKA-AKAP pathway. However, further studies are needed to clarify whether carboplatin phosphorylates TRPA1 using not only TRPA1-expressing HEK293 cells, but also DRGs in carboplatin-induced CIPN model mice.

Carboplatin was developed to overcome cisplatin-induced CIPN and maintain its anticancer efficacy [1]. While most studies suggest that the incidence of CIPN induced by carboplatin is less frequent and less severe than that caused by cisplatin or oxaliplatin, high concentrations of carboplatin cause CIPN in patients receiving multiple drug therapy [1,4,5,6,8,44]. In addition, it has been proposed that high-dose carboplatin shows increased therapeutic effects in cancer patients [2]. Taken together, it might be very important to reveal the pathological mechanisms of CIPN caused by carboplatin using appropriate animal models.

In conclusion, the present study, for the first time, clarified a mechanism of the effect of carboplatin-induced CIPN. Carboplatin induced mechanical allodynia and cold hyperalgesia by enhancing cellular sensitization to TRPA1 through the cAMP-PKA-AKAP pathway. Therefore, modulation of TRPA1 functions might become a target for prevention or therapy of carboplatin-induced CIPN, which could improve chemotherapy with carboplatin to improve their quality of life and prognosis of cancer patients.

## 4. Materials and Methods

### 4.1. Chemicals and Reagent

The following reagents were used in the present study: carboplatin, poly-D-lysine, U73122, KT5720, forskolin, and Hank’s balanced salt solution (Sigma-Aldrich, St Louis, MO, USA); fetal bovine serum (Gibco, Carlsbad, CA, USA); penicillin/streptomycin and AITC (Nacalai Tesque, Kyoto, Japan); blasticidin S and zeocin (Invitrogen, Carlsbad, CA, USA); acetone, 0.25% trypsin-EDTA, Dulbecco’s modified Eagle’s medium and tetracycline (Fujifilm Wako Pure Chemical, Osaka, Japan); fura-2 acetoxymethyl ester (Dojindo Laboratories, Kumamoto, Japan); AKAP I (Tocris Bioscience, Bristol, UK). All other reagents were of the highest purity available from commercial sources.

### 4.2. Experimental Animals

C57BL mice (Clea-Japan, Tokyo, Japan) aged 5 weeks were housed individually in a 12-h light-dark cycle (lights on at 08:00 AM) at a constant temperature and humidity, with ad libitum access to food and water. The mice were acclimatized to laboratory conditions for a week prior to the experiment. All experiments were conducted in accordance with the ethical guidelines of the International Association for the Study of Pain, and approved by the Committee for Ethics of Animal Experimentation of National Cancer Center (Approval Nos. T17–044, 17 May, 2017). During the experiments, efforts were made to minimize the number and suffering of the animals used.

### 4.3. Induction of CIPN Model Mice Using Carboplatin

To establish carboplatin-induced neuropathy models, carboplatin (10 mg/kg) was interperitoneally injected to C57BL mice (male, 6-week-old) twice a week for 2 weeks (Day 0, 3, 7, and 10). A TRPA1 antagonist, HC-030031, or an AKAP inhibitor, AKAP I, was interperitoneally or intrathecally administered to carboplatin-induced neuropathy model mice, respectively.

#### 4.3.1. Behavioral Analysis of Carboplatin-Induced CIPN Model Mice Using the Von Frey Test

Mechanical allodynia was assessed by the von Frey test as previously described [45]. Briefly, mice were placed in a plastic box on a metal grid floor and acclimatized for 60 min prior to testing. Next, von Frey filaments (Muromachi Kikai Co., Ltd., Tokyo, Japan) were applied perpendicularly to the plantar skin of the left hind paw for 2 to 3 s with enough force to slightly bend it. Paw withdrawal responses to the stimulation by von Frey filaments were regarded as a positive response. We evaluated paw withdrawal threshold by using the up-and-down paradigm.

#### 4.3.2. Acetone Test

Cold hyperalgesia was assessed by an acetone test as previously described [22,46]. Briefly, a droplet (100 μL) of acetone, formed on a syringe, was gently touched to the plantar skin of the right hind paw of mice. The time (s) of flinching and biting of the stimulated hind paw was counted for 2 min. The time (s) of paw flinching and biting was reported as the mean of at least two different trials.

#### 4.3.3. Hot Plate Test

A hot plate test was performed to assess thermal hyperalgesia, as previously described [46]. Briefly, mice were placed in a Hot/Cold Plate Analgesia Meter (Muromachi Kikai Co., Ltd., Tokyo, Japan), and the temperature of the plate was maintained at 55 °C. A cut-off time of 30 s was set to prevent tissue damage. The latency of the first hind paw licking or flinching was recorded as paw withdrawal threshold(s). The paw withdrawal threshold was reported as the mean of at least two different trials.

### 4.4. Western Blotting Analysis

Lumbar DRG were isolated from mice and then homogenized in lysis buffer. Next, protein samples were diluted in sodium dodecyl sulfate sample buffer. After heating at 95 °C for 5 min, equal amounts of proteins were separated by sodium dodecyl sulfate-polyacrylamide gel electrophoresis, and then blotted onto nitrocellulose membranes. The membranes were blocked with blocking buffer containing 5% skim milk overnight (16–20 h) at 4 °C and incubated for 2 h at room temperature (22–28 °C) with primary mouse IgG antibodies against GAPDH (1:1000) (Santa Cruz Biotechnology, Inc., Santa Cruz, CA, USA) or primary rabbit IgG antibodies against TRPA1 (1:1000) (Abcam, Cambridge, UK). Next, the membranes were washed and further incubated with horseradish peroxidase-linked anti-mouse IgG antibody (1:1000) (Santa Cruz Biotechnology, Inc.) or anti-rabbit IgG antibody (1:2000) (Southern Biotechnology Associates Inc., Birmingham, AL, USA) for 2 h at room temperature (22–28 °C). Immunoreactivity was detected using a Chemi-Lumi One Super (Nakarai Tesque, Kyoto, Japan). Finally, the band densities of both TRPA1 and GAPDH were measured using a computerized image analysis program (Multi Gauge; Fujifilm, Tokyo, Japan). The level of TRPA1 expression was calculated using the ratio of the TRPA1-specific band density/ the GAPDH-specific band density.

### 4.5. Cell Culture

All in vitro experiments were performed by using HEK293 cells (RIKEN BRC, Ibaraki, Japan) stably expressing hTRPA1, which were prepared by a T-Rex^TM^ system (Invitrogen). The cells were maintained in Dulbecco’s modified Eagle’s medium supplemented with 10% fetal bovine serum, penicillin (100 U/mL), streptomycin (100 mg/mL), Zeocin™ (0.1 mg/mL), and blasticidin S (0.05 mg/mL) at 37 °C in a humidified atmosphere of 95% air and 5% CO_2_. The cells were treated with tetracycline (1 µg/mL) the day before experiment.

### 4.6. Measurement of Intracellular Ca^2+^ Concentrations

The HEK293 cells expressing hTRPA1 were plated on a glass-bottomed dish and then treated with 5 μM fura-2 acetoxymethyl ester in Hank’s balanced salt solution for 20 min at 37 °C. Next, the cells were washed and pretreated with anticancer drugs for the designated duration (10, 30, and 60 min) at room temperature (22–28 °C). Thereafter, these cells were continuously treated with AITC. The fluorescence intensity of the cells was measured at excitation wavelengths of 340 and 380 nm and an emission wavelength of 510 nm. Video image output was digitized by an AquaCosmos 2.5 (Hamamatsu Photonics, Shizuoka, Japan). The extent of the increase in intracellular Ca^2+^ concentration ([Ca^2+^]_i_) induced by AITC was quantified by determining the differences between the ratio (340/380) of the basal and peak levels after AITC treatment.

### 4.7. cAMP Assay

hTRPA1-expressing HEK293 cells were seeded on poly-D-lysine-coated 24-well microplate (Thermo Fisher Scientific, Inc., Waltham, MA, USA) at a density of 1.25 × 10^5^ cells/well. After incubation at 37 °C for 16–24 h, the cells were treated with vehicle or carboplatin (10 µM) for 0, 10, or 30 min in Hank’s balanced salt solution. Next, the cells were washed, mixed with lysis buffer (70 µL/well), and collected by pipetting. The amount of intracellular cAMP was measured using a cAMP-Screen Direct ELISA kit (Applied Biosystems, Waltham, MA, USA) according to the manufacturer’s instructions.

### 4.8. Statistical Analysis

The data are presented as the mean ± standard error of the mean. Statistical analysis was performed using one- or two-way analysis of variance followed by Bonferroni’s multiple comparison test (GraphPad Prism 6; GraphPad Software, San Diego, CA, USA). A probability value (*p*) of <0.05 was considered statistically significant.

## Figures and Tables

**Figure 1 ijms-20-03271-f001:**
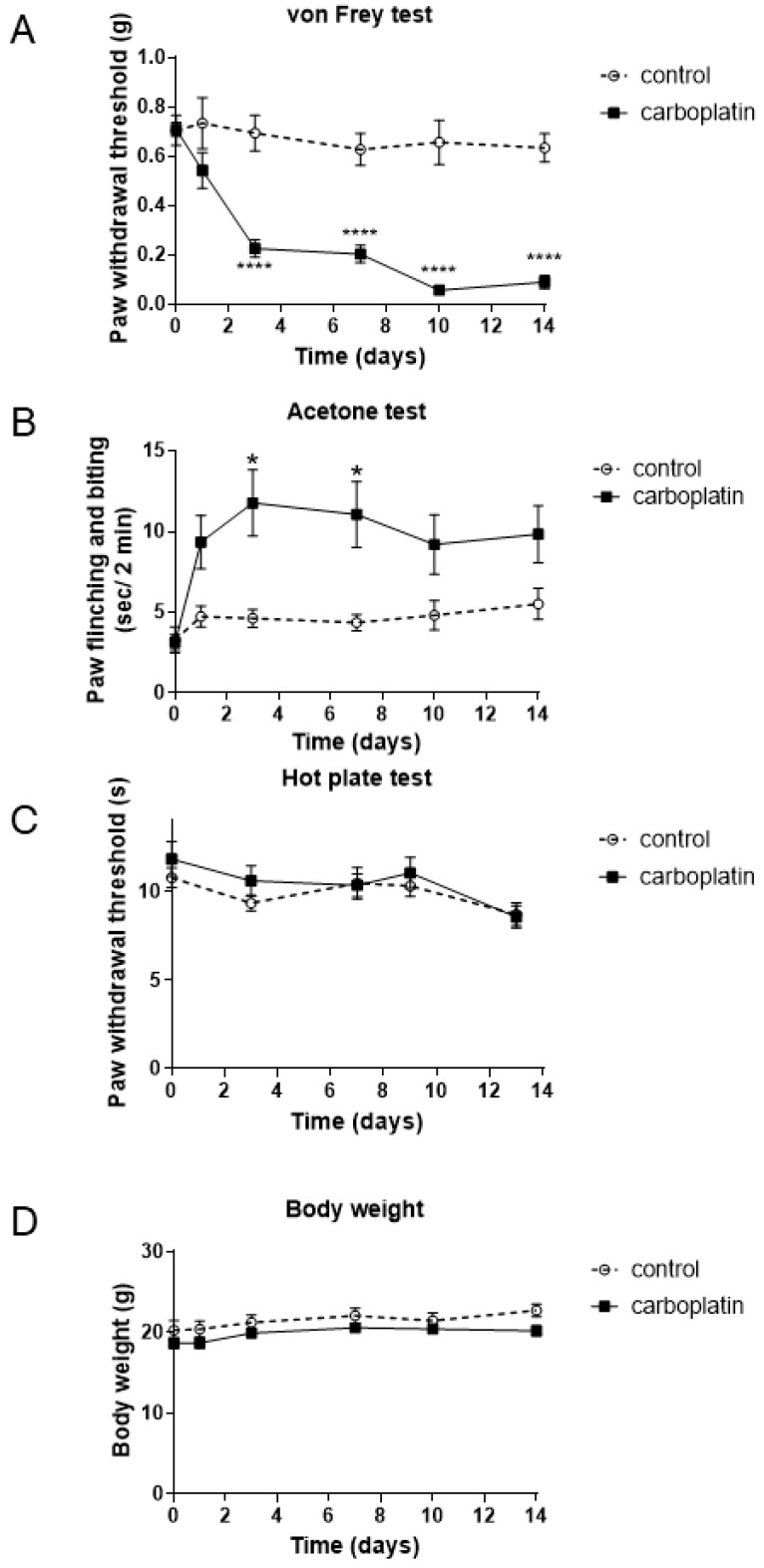
Effects of carboplatin on mechanical allodynia, as well as cold and heat hyperalgesia. Carboplatin (10 mg/kg) or its vehicle (control) was administered intraperitoneally to mice twice a week for 2 weeks. von Frey test (**A**), acetone test (**B**), and hot plate test (**C**) were performed and body weight (**D**) was measured. The data are expressed as the mean ± standard error of the mean (*n* = 5–17). * and **** indicate *p* < 0.05 and *p* < 0.0001, respectively, compared with each control group; Bonferroni’s multiple comparison test following two-way analysis of variance.

**Figure 2 ijms-20-03271-f002:**
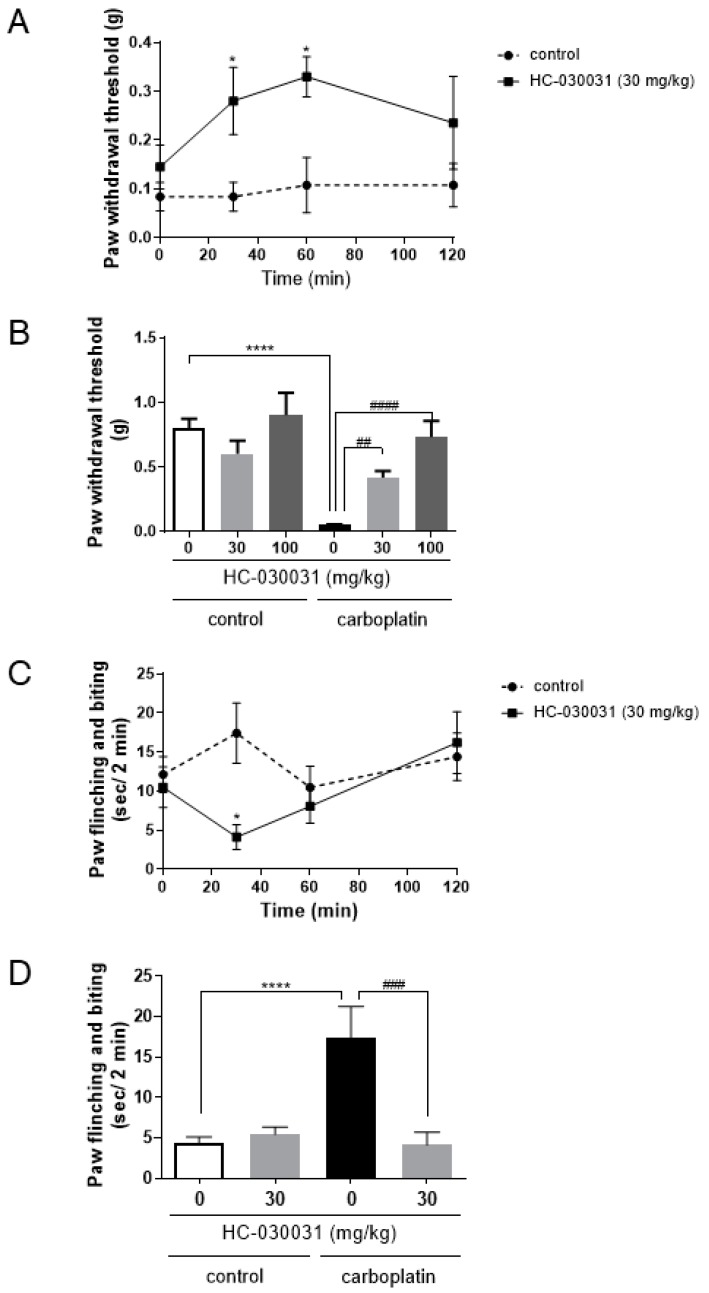
Effects of a TRPA1 inhibitor on carboplatin-induced mechanical allodynia and cold hyperalgesia. HC-030031, a selective TRPA1 antagonist, was administered intraperitoneally to mice at Day 7 after treatment with carboplatin, which were then subjected to von Frey test (**A**,**B**) and acetone test (**C**,**D**). Injection of HC-030031 ameliorated mechanical allodynia in carboplatin-induced peripheral neuropathy model mice in a time (**A**)- and dose (**B**)-dependent manner. At 30 min after injection of HC-030031, carboplatin-induced peripheral neuropathy model mice showed significant amelioration of cold hyperalgesia (**C**,**D**). The data are expressed as the mean ± standard error of the mean (*n* = 4–8). * and **** indicate *p* < 0.05 and 0.0001, respectively, compared with the control group; ##, #### indicate *p* < 0.01 and 0.0001, respectively, compared with the carboplatin group; Bonferroni’s multiple comparison test following one (**B**,**D**)- or two (**A**,**C**)-way analysis of variance.

**Figure 3 ijms-20-03271-f003:**
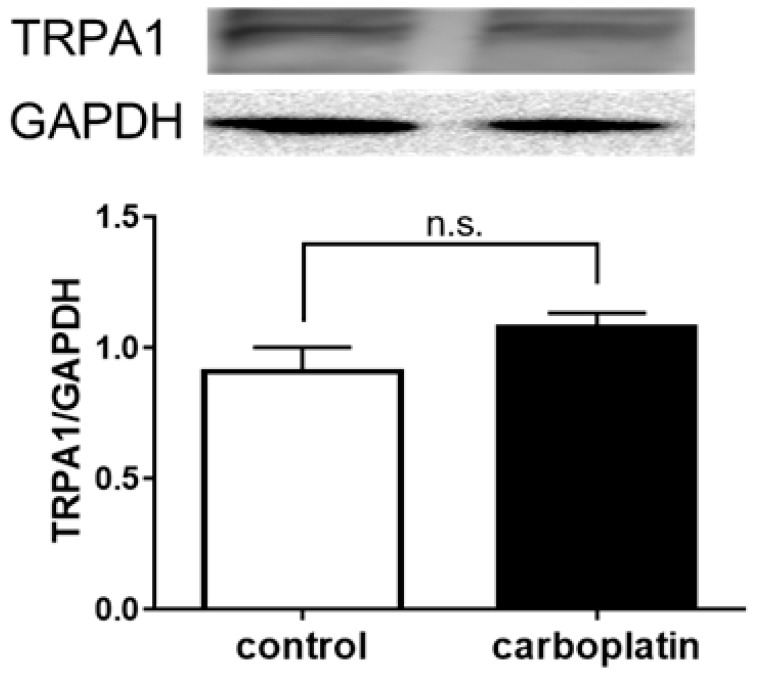
Effects of carboplatin on the expression of TRPA1 protein in mouse DRG. The protein expression of TRPA1 in the DRG of carboplatin-induced peripheral neuropathy model mice was measured by western blotting. The data are expressed as the mean ± standard error of the mean (*n* = 5–7): compared with the control group; Bonferroni’s multiple comparison test following one-way analysis of variance. n.s.; not significant.

**Figure 4 ijms-20-03271-f004:**
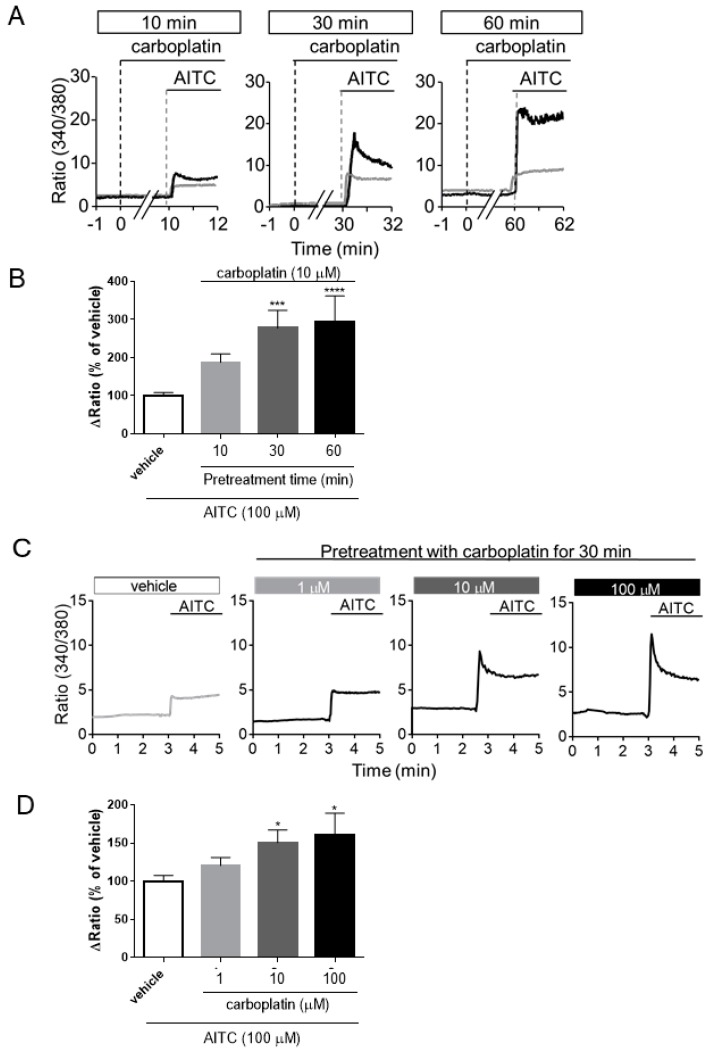
Carboplatin enhances TRPA1 activation in a time- and dose-dependent manner in TRPA1-expressing HEK293 cells. Fura 2-loaded, hTRPA1-expressing cells were pretreated with vehicle (gray line) or carboplatin (10 µM, black line) for 10, 30, or 60 min and then treated with allyl isothiocyanate (AITC) (100 µM), respectively (**A**,**B**). Concentration-response relationship of carboplatin and increases in [Ca^2+^]_i_ induced by AITC, respectively (**C**,**D**). Representative tracing of the mean [Ca^2+^]_i_ in randomly selected cells expressing hTRPA1 (**A**,**C**). The extent of increases in [Ca^2+^]_i_ induced by AITC was quantified by determining the differences between the ratio (340/380) of the basal and peak levels after treatment with each agonist (**B**,**D**). The data are expressed as the means ± standard error of the mean of at least triplicate experiments (*n* = 107–313). *, ***, and **** indicate *p* < 0.05, 0.01, and 0.0001, respectively, compared with the AITC alone group; Bonferroni’s comparison test following one-way analysis of variance.

**Figure 5 ijms-20-03271-f005:**
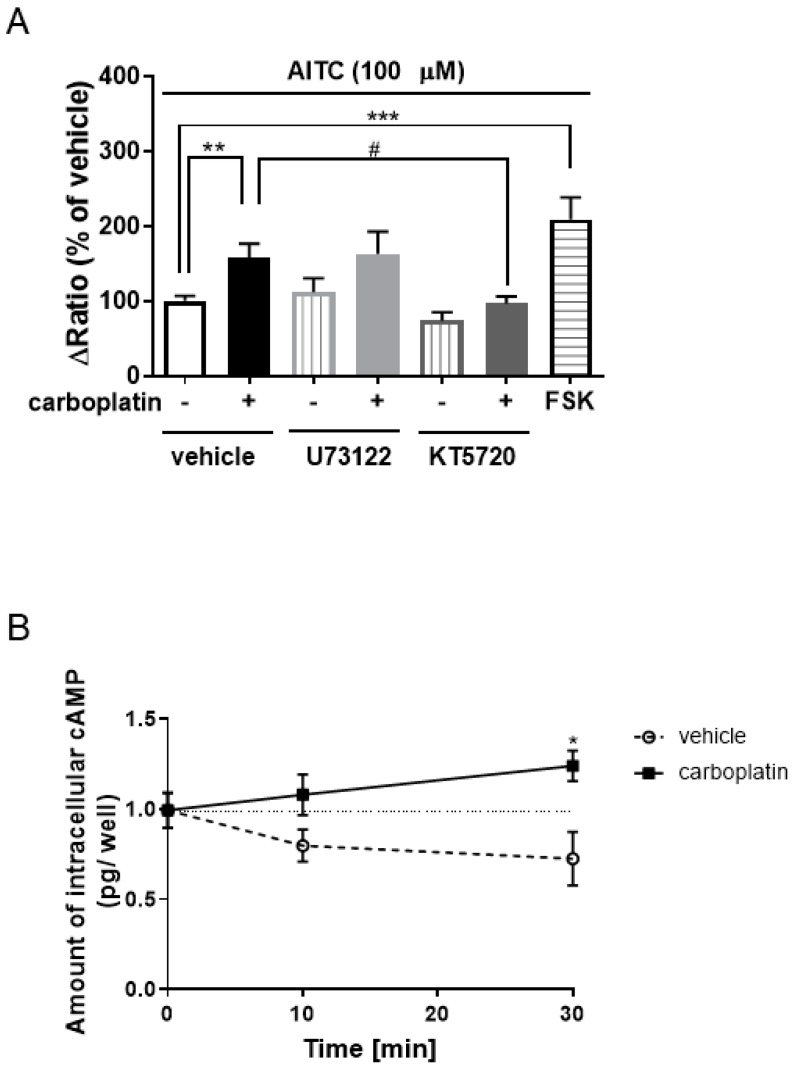
Effects of inhibitors of phospholipase C (PLC) or protein kinase A (PKA) on carboplatin-induced enhancement of TRPA1 activation. (**A**) Fura 2-loaded cells expressing hTRPA1 were pretreated with both carboplatin (10 µM) and each inhibitor [the PLC inhibitor U73122 (1 µM) or the PKA inhibitor KT5720 (0.1 µM)] for 30 min and then treated with allyl isothiocyanate (AITC) (100 µM), respectively. The cells were pretreated with forskolin (FSK; 10 µM), a PKA activator, for 30 min, and treated with AITC (**A**). The extent of increases in [Ca^2+^]_i_ induced by AITC was quantified by determining the differences between the ratio (340/380) of the basal and peak levels after treatment with each agonist (**A**). (**B**) hTRPA1-expressing cells were treated with vehicle or carboplatin (10 µM) for 0, 10, or 30 min, and the amount of intracellular cAMP was measured using cAMP-Screen Direct^®^ systems. The data are expressed as the means ± standard error of the mean of at least triplicate experiments (A, *n* = 68–474; B, *n* = 3–4). *, **, and *** indicate *p* < 0.05, 0.01, and 0.001, respectively, compared with the vehicle group; Bonferroni’s multiple comparison test following one-way analysis of variance (ANOVA). # indicates *p* < 0.05, compared with the carboplatin alone group; Bonferroni’s multiple comparison test following one-way ANOVA.

**Figure 6 ijms-20-03271-f006:**
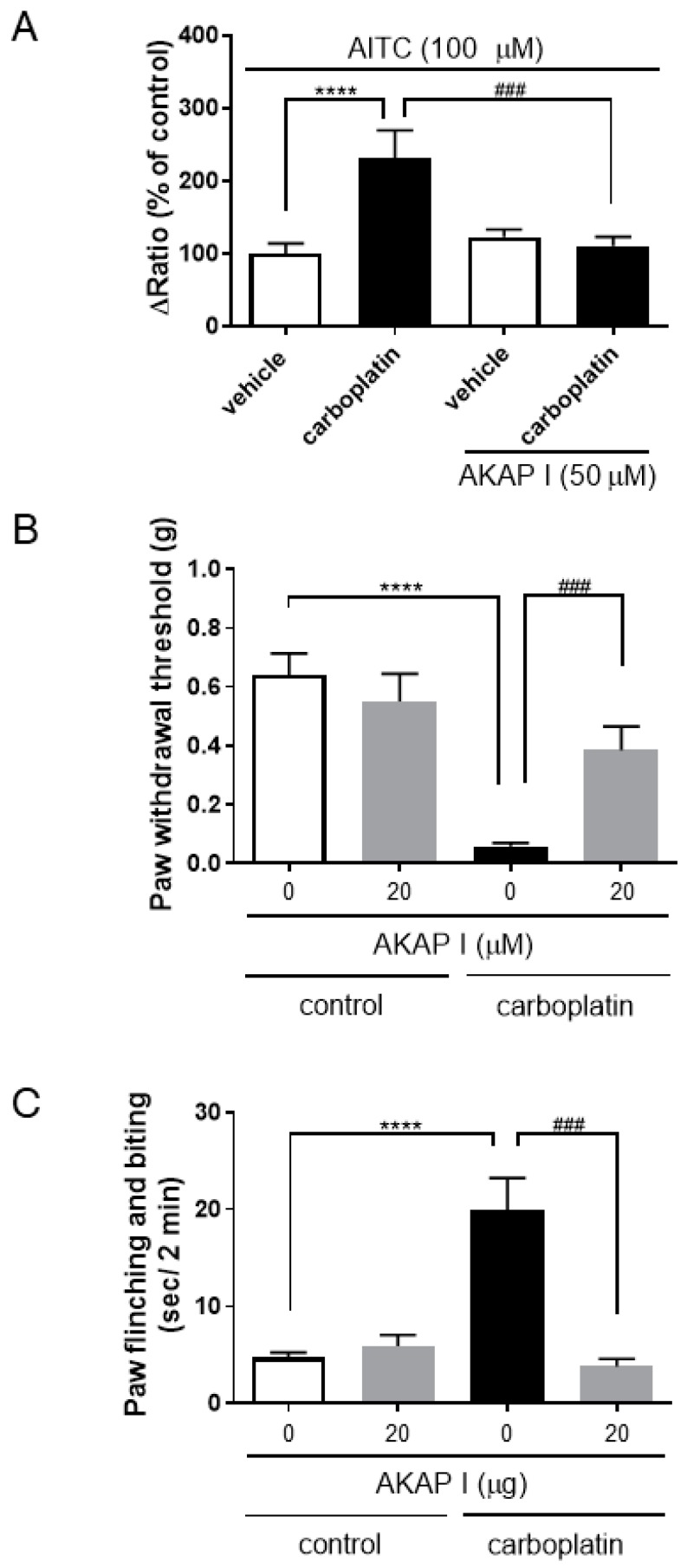
Effects of an inhibitor of A-kinase anchoring protein (AKAP) on TRPA1 activation or mechanical allodynia induced by carboplatin. (**A**) Fura 2-loaded cells expressing hTRPA1 were pretreated with both carboplatin (10 µM) and AKAP St-Ht31 inhibitor peptide (AKAP I; 50 µM) for 30 min and then treated with AITC (100 µM). The extent of increases in [Ca^2+^]_i_ induced by AITC was quantified by determining the differences between the ratio (340/380) of the basal and peak levels after treatment with each agonist (**A**). (**B**) The mice were intrathecally administered AKAP I (20 µg), and then subjected to von Frey test (**B**) and acetone test (**C**). At 60 min after injection of AKAP I (Day 8 and 9), mechanical hyperalgesia in model rats of carboplatin-induced peripheral neuropathy was ameliorated (**B**). At 30 min after injection of AKAP I, there was a significant amelioration of cold hyperalgesia in the model rats (**C**). The data are expressed as the mean ± standard error of the mean of at least triplicate experiments (**A**, *n* = 74–173; **B**, *n* = 4–11; **C**, *n* = 4–14). **** indicates *p* < 0.0001, compared with the vehicle or control group; #### indicates *p* < 0.001, compared with the carboplatin alone group; Bonferroni’s multiple comparison test following one -way analysis of variance.

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
