# Peer review of "Carboplatin Enhances the Activity of Human Transient Receptor Potential Ankyrin 1 through the Cyclic AMP-Protein Kinase A-A-Kinase Anchoring Protein (AKAP) Pathways"

_ijms, 2019, doi:10.3390/ijms20133271_

Reviewer 1 Report

The manuscript by Miyano et al shows that carboplatin, an anticancer drug, induces mechanical allodynia and cold hyperalgesia by enhancing the sensitization of the human transient receptor potential ankyrin 1 (TRPA1) through the cAMP-PKA-AKAP pathway. It is an interesting study that aims to better understand the molecular pathways responsible for the peripheral neuropathy induced by chemotherapy, which is an important topic to eventually improve the well-being of patients.

Overall, the experiments are appropriately designed and conducted, and the results obtained are very interesting. Consequently, I have only some minor suggestions to improve the manuscript:

In figure 2, it is not clear how long was the carboplatin treatment before the TRPA1 antagonist administration? In panels A and C, the “control” condition should be “carboplatin” instead.

Figure 3: Previous reports have demonstrated that TRPA1 sensitization can be regulated by phosphorylation by PKA (ref: Meents et al., 2017, PLosOne 12(1):e0170097). In addition to assess TRPA1 expression levels, it could be interesting to determine its phosphorylation status in mouse DRG.

In figure 6, panel B, “AKAP” units should be µg instead of µM, as stated in the figure legend.

Author Response

Reply to the comments Reviewer #1:

We thank the Reviewer #1 for a very careful review of our manuscript and for providing constructive comments. We have considered each of the suggestions in turn and have reviewed some articles quoted in the revised manuscript to improve our paper according to reviewer’s comments.

Comments of Reviewer #1:

1.    In figure 2, it is not clear how long was the carboplatin treatment before the TRPA1 antagonist administration? In panels A and C, the “control” condition should be “carboplatin” instead.

Response: We thank the reviewer for the constructive comment. We administrated TRPA1 antagonist to mice at Day7 after treatment with carboplatin. Therefore, we have added its associated description in the “Figure Legends”section (Revised version) as follows;

Page 4, Line 98-100; HC-030031, a selective TRPA1 antagonist, was administered intraperitoneally to mice at Day 7 after treatment with carboplatin, which were then subjected to von Frey test (A, B) and acetone test (C, D).

In addition, we have redrawn “Figures 2A and 2C”(Previous version) in response to the comment of the reviewer.

2.    Figure 3: Previous reports have demonstrated that TRPA1 sensitization can be regulated by phosphorylation by PKA (ref: Meents et al., 2017, PLosOne 12(1):e0170097). In addition to assess TRPA1 expression levels, it could be interesting to determine its phosphorylation status in mouse DRG.

Response: We thank the reviewer’s valuable comments. It is very important to analyze whether the level of TRPA1 phosphorylation is increased in mouse DRG treated with carboplatin. In response to the comment of the reviewer, we have rearranged the sentences to describe its importance in the Discussion section (Revised version) as follows;

Page 10, Line237-239; However, further studies are needed to clarify whether carboplatin phosphorylates TRPA1 using not only TRPA1-expressing HEK293 but also mice DRGs in carboplatin-induced CIPN model mice.

3.    In figure 6, panel B, “AKAP” units should be µg instead of µM, as stated in the figure legend.

Response: We are grateful to the reviewer for pointing out the error. We have redrawn Figures 6B (Previous version).

Reviewer 2 Report

Miyano et al. report that carboplatin enhances the activity of human TRPA1 ion channel through cAMP-PKA-AKAP pathway. Further, the authors show that carboplatin sensitizes TRPA1 in vivo rather than alters its expression. This study is well performed with proper tool compounds, statistical analysis and controls.

Author Response

Reply to the comments Reviewer #2:

We are grateful for the reviewer’s comments on the manuscript. We will make further efforts in writing more good papers.